# Deferasirox-Dependent Iron Chelation Enhances Mitochondrial Dysfunction and Restores p53 Signaling by Stabilization of p53 Family Members in Leukemic Cells

**DOI:** 10.3390/ijms21207674

**Published:** 2020-10-16

**Authors:** Chiara Calabrese, Cristina Panuzzo, Serena Stanga, Giacomo Andreani, Silvia Ravera, Alessandro Maglione, Lucrezia Pironi, Jessica Petiti, Muhammad Shahzad Ali, Patrizia Scaravaglio, Francesca Napoli, Carmen Fava, Marco De Gobbi, Francesco Frassoni, Giuseppe Saglio, Enrico Bracco, Barbara Pergolizzi, Daniela Cilloni

**Affiliations:** 1Department of Clinical and Biological Sciences, University of Turin, 10043 Turin, Italy; k.kalabrese81@gmail.com (C.C.); giacomo.andreani@unito.it (G.A.); alessandro.maglione@unito.it (A.M.); lucrezia.pironi@edu.unito.it (L.P.); jessica.petiti@unito.it (J.P.); muhammad.ali79@edu.unito.it (M.S.A.); patrizia.scaravaglio@unito.it (P.S.); carmen.fava@unito.it (C.F.); marco.degobbi@unito.it (M.D.G.); francescofrassoni@gaslini.org (F.F.); giuseppe.saglio@unito.it (G.S.); barbara.pergolizzi@unito.it (B.P.); daniela.cilloni@unito.it (D.C.); 2Department of Neuroscience Rita Levi Montalcini, Neuroscience Institute Cavalieri Ottolenghi, University of Turin, 10126 Turin, Italy; serena.stanga@unito.it; 3Human Anatomy Section, Department of Experimental Medicine, University of Genoa, 16132 Genova, Italy; silvia.ravera@unige.it; 4Department of Oncology, University of Turin, 10043 Turin, Italy; francesca.napoli@unito.it (F.N.); enrico.bracco@unito.it (E.B.)

**Keywords:** iron, Deferasirox, chelation, leukemia, mitochondria, p21, p53, p73, MDM2

## Abstract

Iron is crucial to satisfy several mitochondrial functions including energy metabolism and oxidative phosphorylation. Patients affected by Myelodysplastic Syndromes (MDS) and acute myeloid leukemia (AML) are frequently characterized by iron overload (IOL), due to continuous red blood cell (RBC) transfusions. This event impacts the overall survival (OS) and it is associated with increased mortality in lower-risk MDS patients. Accordingly, the oral iron chelator Deferasirox (DFX) has been reported to improve the OS and delay leukemic transformation. However, the molecular players and the biological mechanisms laying behind remain currently mostly undefined. The aim of this study has been to investigate the potential anti-leukemic effect of DFX, by functionally and molecularly analyzing its effects in three different leukemia cell lines, harboring or not p53 mutations, and in human primary cells derived from 15 MDS/AML patients. Our findings indicated that DFX can lead to apoptosis, impairment of cell growth only in a context of IOL, and can induce a significant alteration of mitochondria network, with a sharp reduction in mitochondrial activity. Moreover, through a remarkable reduction of Murine Double Minute 2 (MDM2), known to regulate the stability of p53 and p73 proteins, we observed an enhancement of p53 transcriptional activity after DFX. Interestingly, this iron depletion-triggered signaling is enabled by p73, in the absence of p53, or in the presence of a p53 mutant form. In conclusion, we propose a mechanism by which the increased p53 family transcriptional activity and protein stability could explain the potential benefits of iron chelation therapy in terms of improving OS and delaying leukemic transformation.

## 1. Introduction

Iron is a cofactor for many biochemical processes including oxidative phosphorylation, oxygen storage, and enzymatic reactions, thus it is essential for the survival of nearly all types of cells. In particular, it has been proven that its depletion leads to impaired proliferation, growth, and survival of cancer cell lines [1].

Iron excess, through the generation of intracellular Reactive Oxygen Species (ROS), increases the level of genomic instability and lipid peroxidation and is implicated in several types of cancer [2,3], accelerating disease progression [4]. In this scenario, iron deprivation showed beneficial effects on tumors cells in terms of apoptosis and cell cycle arrest [5,6].

Patients affected by Myelodysplastic Syndromes (MDS) and acute myeloid leukemia (AML) are generally characterized by iron overload (IOL) due to periodical red blood cell (RBC) transfusions. Moreover MDS patients are prone to iron loading, even in absence of RBC transfusion [7]. Several studies indicate that IOL causes organ damage and eventually impacts on overall survival (OS) [8]. Furthermore, recent studies have reported potential effects of iron chelation therapy (ICT) on leukemic patients’ OS, however these evidences have not been observed in all the cohorts [9,10,11,12,13]. Experimental data generated in a MDS murine model with IOL indicated that iron chelation affects the disease progression [14]. In this regard, in MDS patients, the oral iron chelator Deferasirox (DFX) improves the OS and delays leukemic transformation [13,15,16]. Though alteration of various biological processes have been associated to DFX-dependent iron chelation (including decreased oxidative DNA damage [17], induction of differentiation of leukemia blasts and anti-proliferative action through inhibition of the mTORC1 pathway [18], the molecular mechanisms underlying its activity are not well defined yet. We have also previously reported that DFX is a potent inhibitor of nuclear factor kappa B (NF-κB) in MDS and AML cells [19]. Interestingly, p53 is known to play a pivotal role in controlling iron homeostasis by direct interaction with Ferredoxin Reductase (FDXR) and Iron Responsive Element Binding Protein 2 (IRP2) proteins [20,21,22]. Additionally, iron depletion triggered p53 phosphorylation and stabilization, thus preventing its proteasomal degradation [23,24,25,26]. Recently, Shen and colleagues demonstrated that intracellular iron overload shortens p53 half-life by promoting p53 nuclear export and cytosolic degradation [27,28,29]. These data provide insights into tumorigenesis associated to iron excess, suggesting that the p53 family might represent an interesting target to be investigated during iron chelation therapy. In addition, restoration of p53 pathway signaling via its homologs p63 and p73 has been described in several tumor models exhibiting p53 mutant forms [30,31]. Given the high structural similarity among the family members (until 70% in the DNA-binding domain), they can bind the same responsive promoters [32,33,34,35]. Thus, they share overlapping functions due to their ability to trans-activate the same genes like cyclin-dependent kinase (CDK) inhibitor (*CDKN1A* (p21)), Bcl2 Binding Component 3 (*PUMA*), and Murine Double Minute 2 (*MDM2*) [36,37,38].

p21 is a potent CDK inhibitor that regulates cell cycle progression at the checkpoint level between G1 and S phases, leading to arrest in G1 phase in response to different stressors [32,39]. PUMA is a direct regulator of apoptosis that acts by inducing cytochrome c release and rapid induction of programmed cell death [40]. MDM2 is an E3 ubiquitin-protein ligase which plays a central role in regulating the stability of p53 family proteins. It modulates p53 and p73 activity by inducing its degradation through ubiquitination [41]. Nevertheless, p53 family regulates the activity and the production of MDM2 both directly and indirectly, resulting in a negative self-regulatory feedback [42,43,44].

For all the above reasons, we decided to investigate a putative contribution of p53 family member proteins to the potential anti-leukemic effect of iron deprivation on MDS and AML cells, especially since p53 is frequently mutated or deleted in these pathologies [45,46], and it is associated with leukemia onset and relapse [47,48,49,50,51].

Our findings suggested that treatment with DFX is crucial to modulate viability, growth arrest, and apoptosis of leukemia cell lines and MDS/AML primary cells. Moreover, our results imply that DFX acts by increasing p53 and p73 stability. The latter, in case of p53 deficiency, could have a compensatory role by activating the same p53 pathways. Finally, we observe that mitochondrial shape, network, and activity, regulated even by p53 family activity [52,53], are severely sensitive to iron level, suggesting that iron deprivation condition, via mitochondria impairment, could be essential to regulate leukemic cells’ features.

## 2. Results

### 2.1. Iron Chelation Induces the Fragmentation of Mitochondrial Network and a Dysfunction in the Oxidative Phosphorylation in Acute Myeloid Leukemia Cell Lines

To investigate if iron deprivation could alter the mitochondria network even in leukemia, we decided to treat three different AML cell lines, HL60, NB4, and MOLM-13, with 50 µM of DFX for 48h and then to examine mitochondria dynamics. The analysis of the mitochondria shapes was performed by using the MiNA toolset [54], which allows to obtain parameters to quantitatively capture the morphology of the mitochondrial network. In all cell lines under physiological conditions, mitochondria were interconnected, thus forming an intracellular network (Figure 1A). By contrast, when compared to the respective controls, DFX treatment caused a severe alteration of the mitochondrial network, as highlighted by the skeletal images and by the mitochondria footprint quantification (Figure 1B). Indeed, DFX-treated cells showed a higher mitochondrial area compared to non-treated cells, while the mean mitochondria branch length is shorter in treated cells. Both morphological information clearly indicated an alteration of mitochondrial dynamics, in turn affecting the mitochondrial network and potentially, the function.

These morphological alterations seem to be associated with a dysfunctional activity of the oxidative phosphorylation (OxPhos). In particular, all myeloid leukemia cell lines treated with DFX displayed a reduction of oxygen consumption rate (OCR) and ATP synthesis through F0-F1 ATP synthase (Table 1). Moreover, the residual OxPhos activity appeared less efficient since the P/O value is lower with respect to those observed in the untreated sample (Table 1).

In addition, as a consequence of this metabolic alteration, the ATP/AMP ratio, a marker of cellular energy metabolism, was very low in the DFX-treated samples in comparison to the controls, due to a decrement of ATP and an increment of AMP intracellular levels (Table 2). Subsequently, investigating by Western blot, the expression of Aconitase 2, a crucial mitochondrial protein, we observed a significant reduction (Appendix A), proving the importance of mitochondrial morphogenesis machinery components on mitochondrial function and activity. All these results demonstrate that iron chelation affects mitochondria in leukemic cells, representing a possible attractive anti-leukemic strategy.

### 2.2. Deferasirox Exerts In Vitro Anti-Leukemic Activity on Acute Myeloid Leukemia Cell Lines and on Primary MDS Cells

To assess whether the morphological alterations and dysfunctional activity of mitochondria were associated to cell viability, we analyzed the rate of cell proliferation and apoptosis after iron chelation treatment. Initially, we treated MOLM-13, HL60, and NB4 cell lines with 10, 25, 50, or 100 µM of DFX for 48 h. DFX inhibited the growth of all tested cell lines in a dose-dependent manner, reaching the maximum effect at 100 µM, with a reduction in proliferation close to 70% in all cells tested (Figure 2A). These promising results were confirmed by viability assay performed by Fluorescence Activated Cell Sorting (FACS) (Appendix A), with a significant dose-dependent reduction of viability after DFX exposition when compared to untreated cells. To validate our experimental protocol, we investigated iron level after 48 h of DFX treatment, at concentration of 25 and 50 µM, that appeared to be the most tolerated concentrations, also capable to heavily decrease the percentage of proliferation, using the calcein fluorescence assay. In the presence of the chelator, a dose-dependent increase in intracellular calcein fluorescence signal confirmed the reduction in the content of labile iron pool (LIP), validating our experimental tools and suggesting a clear relationship between iron chelation and anti-proliferative effects in acute leukemia cells (Figure 2B and Appendix A). Consistently, we also noticed an induction of apoptosis. In more detail, flow cytometric evaluation of Annexin V/PI-stained cells demonstrated a significant induction of cell death in all the tested cell lines, with a percentage of apoptotic cells reaching 20–30% at 50 µM (Figure 2C). As shown in Figure 2D, cleaved caspase-3 levels, analyzed by Western blot, increased dramatically in DFX-treated cells, according to the drug concentration.

We next attempted to validate these data in MDS/AML primary cell cultures. Primary cells from 5 patients at diagnosis belonging to the different World Health Organization (WHO) categories, whose clinical and cytogenetic features are illustrated in Table 3 (Sample 1–5), were used for proliferation and apoptosis assays. Five healthy subjects were already used as controls and their characteristics are reported in Table 3. Interestingly, ferritin levels, which is a standard indirect parameter of iron content, were totally different from those of MDS, confirming that the iron content of WBC isolated from normal samples was extremely low. In this regard, DFX inhibited the vitality of MDS primary cells, reaching the maximum effect at 100 µM, by a reduction close to 50% (Figure 2E). At the same time, we did not observe a significant reduction of viability on healthy subjects’ specimens. This effect was confirmed by monitoring apoptotis in MDS and healthy cells by flow cytometry (Figure 2F). All these results suggested a clear relationship between iron chelation and anti-proliferative/pro-apoptotic effects in leukemia cells. The lack of effect on healthy donors’ cells increased the relevance of our work and suggested that iron overload may represent a new target to exploit by iron chelation, to obtain a specific anti-leukemic effect.

### 2.3. Deferasirox Activates p53 Targets on Acute Myeloid Leukemia Cell Lines and on Primary MDS/AML Cells

Based on the obtained results, we decided to investigate a possible involvement of p53 on the mitochondrial alterations and on the anti-leukemic effects observed. Our hypotheses were supported by data already reported, indicating that p53 is able to move to mitochondria and to induce caspase activation during the apoptosis process [56]. In addition, p53 can play a role in mitochondrial dynamics by regulating genes such as *DRP1* [57]. Furthermore, it is known that iron depletion increases p53 protein amount by preventing its proteasomal degradation [27]. To investigate if DFX exerts its activity on acute leukemia samples in a p53-dependent manner, we analyzed the expression of well-known p53 target genes in our acute myeloid leukemia cell lines that are indeed characterized by a peculiar p53 genotype. Indeed, MOLM cells harbor a wild-type p53 locus, whereas HL-60 and NB4 are p53 null and p53 R248Q, respectively. Treatment with DFX induced a sharp increase in the expression of cell cycle-dependent kinase inhibitor *CDKN1A* (p21) and of the pro-apoptotic *PUMA* genes (Figure 3A), corroborating previous results. Surprisingly, their transcriptional regulation was sensitive to DFX treatment independently from the p53 mutational status, suggesting that DFX effects could depend from a p53 family member other than p53 itself. Instead, *GADD45* did not appear to be significantly modified after iron chelation. Thus, it is plausible as a real implication of the p53 family, even if we do not exclude the existence of other mechanisms involved. Moreover, a marked increase in p21 protein was observed in all DFX-treated cells (Figure 3B), hence confirming the gene expression results. On the contrary, we noticed a discrepancy for MDM2 protein. In more detail, with some minor differences between the tested cell lines, we observed that DFX increased the mRNA (Figure 3A) but lowered the protein level of MDM2 compared to untreated cells (Figure 3C). Nevertheless, these results confirmed the dynamic p53-MDM2 negative feedback loop. Subsequently, primary cells of 15 MDS/AML patients, whose clinical and cytogenetic features are illustrated in Table 3, were used to evaluate the effect of DFX on the expression of the *CDKN1A* and *PUMA* genes. Like what we observed in leukemia cell lines, the *CDKN1A* and *PUMA* gene expression significantly increased after DFX incubation (Figure 3D,E). Therefore, these data led us to propose that also in MDS/AML primary cells, DFX could activate specific p53-depepndent gene transcription. 

### 2.4. Deferasirox Regulates p53 and p73 Protein Stability

The p53 family includes two additional closely related members, p63 and p73. They share a high degree of structural homology with p53 and can activate the transcription of most p53-sensitive genes. In addition, the stability of both p53 and p73 is tightly controlled by the ubiquitin-proteasome system through the MDM2 E3 ubiquitin-ligase [58]. For that reason, and in order to identify a plausible explanation for the unexpected gene expression pattern observed even in HL60 and NB4 p53-deficient cell lines, we examined the p53 and p73 protein levels, and their subcellular localization, in the three cell lines after treatment with 50 µM Deferasirox. The results obtained by immunofluorescence showed that DFX increased p73 levels by inducing a sharp nuclear protein accumulation (Figure 4A). p53 also appears to be increased, even if with less intensity, except for the p53 null HL60 cell line. These results were further confirmed by determining the levels of cellular fluorescence from fluorescence microscopy images (Appendix A), and by analyzing total protein levels by Western blot (Appendix A).

We next moved to investigate p53 and p73 proteins by IHC in 5 BM MDS patients at diagnosis (DX) and within one year of iron chelation treatment (ICT) with DFX. Despite the paucity of the cohort, 80% of patients (even if with different intensity among them) displayed a behavior similar to that observed in cell lines culturing. p73 signal significantly increased (Figure 4B and Appendix A) after treatment. In our specimens, p53 signal resulted difficult to detect. We have assumed that since it is extremely crucial to several cellular processes, therefore, even small changes in its amount are enough to activate a specific response. 

To further strengthen our results, we investigated the effect of iron chelation on the transcriptional levels, by analyzing a microarray dataset from the p53-truncated K562 leukemia cell line treated with DFX [18]. Then, we performed a pathway enrichment analysis by using the 258 genes emerged as differentially expressed. Surprisingly, although the cell line analyzed was p53-deficient, the gene ontology analysis performed highlighted an enrichment related to effectors of p53, p73, and p63 networks arisen among the most significant enriched clusters (Figure 4C). Since most of the p53 direct outputs are common to other family members [34,35], our findings indicate that iron depletion-triggered signaling, in the absence of p53, is overtaken by other family members (i.e., p73 and p63). This conclusion was significant for our previous findings concerning p53 family activation, reinforcing our hypothesis. Indeed, *TP53INP1* resulted as a highly expressed gene after iron chelation treatment. Several studies confirmed the ability of p73 to induce *TP53INP1* expression in p53-deficient cells, by enhancing the capacity of p73 to regulate cell cycle progression and apoptosis, regardless of p53 [59]. *PMAIP1*, also known as NOXA, contributes to p53 family-dependent apoptosis by a direct action on MCL1 and subsequent activation of mitochondrial membrane changes. Finally, this analysis revealed that *CDKN1A* was commonly activated by all p53 family proteins after DFX treatment, just like our experimental results. Finally, in Figure 4D, we resumed our hypothesis about the strategic role of iron chelation on mitochondrial activity and on p53 family stability, in order to propose new attractive targets to investigate in DFX-treated patients.

## 3. Discussion

Iron is essential for the normal function of almost all cell types. The requirement of iron is even higher in cancer cells because of their rapid cell growth and proliferation. Previous studies showed that the administration of iron in mice can induce tumor generation [3]. By contrast, several epidemiological studies demonstrated that a reduced incidence of cancer is associated with iron deficiency [60]. In line with these reports, recent studies described the improved leukemia-free survival in patients affected by AML and MDS treated with iron chelation [9,10,11,12,13]. Moreover, the iron overload has been shown to exert a negative impact on the overall survival of patients affected by acute leukemia treated with hematopoietic stem cell transplantation (HSCT). Increased levels of NTBI (non-transferrin-bound iron) and LPI, during hematopoietic stem cell transplantation, have been correlated with poor survival [61,62].

Therefore, following the first clinical case of an acute leukemia patient who achieved complete remission with iron chelation [16], several studies addressed the problem of the anti-leukemic properties of iron chelation [9,10,11]. The results obtained are divergent in some respects, but the ability of chelation therapy to delay leukemic progression seems confirmed. The biological mechanisms responsible for this event are still fairly obscure. We previously reported that Deferasirox can inhibit NF-kB [19], and Ohyashiki et al. demonstrated the mTORC1 pathway inhibition by Deferasirox [18]. More recently, Shen and colleagues [27] provided a new perspective by demonstrating that iron overload can reduce p53 activity and that chelation can stabilize p53, and related the p63 and p73 proteins [63]. They suggested a biological rationale for the relationship between iron overload and increased risk of cancer. It also changed the idea that iron overload reduces survival only by inducing organ damage. Finally, these data could justify one potential anti-leukemic effect of iron chelation anti-leukemic effect, through the reactivation p53 family proteins.

In the present study, we clearly demonstrated that iron chelation triggers apoptosis in leukemic blasts and cell lines and impairs cell growth by enhancing the p53 transcriptional activity. *CDKN1A* and *PUMA*, master regulators of proliferation and apoptosis respectively, represent the most activated genes, corroborating our functional assays. The regulation of p21 by iron chelation has already been reported [39,64]. Our study confirms that p21 expression occurs independently from p53 status, suggesting that stress signals, including iron chelation, could act though p53-related proteins p63 or p73, thus compensating the lack of p53 function. Similar results were obtained by analyzing a microarray dataset concerning a p53-truncated leukemia cell line treated with DFX. The p53 family protein network is heavily enriched, further indicating that the signal triggered by iron depletion, in the absence of p53, is overtaken by the other family members.

To better clarify the mechanism of p53 family reactivation, we investigated the MDM2 status. After cellular stress, p53-enhanced expression is achieved though different mechanisms, including phosphorylation, acetylation, and methylation, which reduced its interaction with MDM2, thus favoring p53 activity [26,65]. Moreover, hypoxia-inducible factor 1 alpha (HIF1a) enforced p53 protein stability [66]. Within this scenario, our attention was focused on MDM2 protein that tightly controlled the stability of p53 and p73 by the ubiquitin-proteasome system [41]. Iron-dependent regulation of MDM2 influenced p53 activity in hepatic cancer cells [67]. In addition, a negative feedback between the two proteins governed the p53 response to cellular stress [42]. In this study, we demonstrated that MDM2 protein, but not mRNA, decreased after iron chelation, thus inducing a stabilization of p53 and p73, as suggested by their sharp nuclear accumulation after DFX treatment. The fact that DFX exerts an activity on MDM2 seems very attractive in view of an anti-leukemic property. The importance of MDM2 inhibition in the leukemic setting is suggested by the clinical results obtained by a small molecule designed to block MDM2 with the aim of increasing p53 levels, named Idasanutlin [68]. Our findings revealed a direct link between iron and stability/functions of p53 family members, providing a new fascinating opportunity for cancer treatment based on iron deprivation. 

Additionally, MDM2 is a potential binding partner able to directly link and block complexes composed of mutant R175H p53 and p73, thus resulting in a loss of the functional wild-type p73 [69,70]. Hence, reduction of MDM2 after iron chelation could further promote p73 transcriptional activity, by reducing its inhibitory ability on mutant p53-p73 complex. 

Moreover, the NB4 cell line that we used is characterized by R248Q mutation, classified as a DNA binding mutant, which differed from other structurally unfolded mutants. Even if both types of mutants have been shown to inhibit p63 and p73 function, the unfolded p53 mutants seem to be more severe and capable to interact with p63 and p73 more heavily than DNA binding mutants [71]. Consequently, the GOF of mutant p53 related to p63 and p73 inhibition could be less pronounced in our p53 mutated cell line model. Furthermore, we demonstrated that iron chelation drastically increased the level of p73 itself, and this phenomenon may further reduce the inhibitory binding effect of mutant p53 on p73. 

Finally, our results suggested that the p53 pathway activated by DFX is strongly connected to mitochondrial damage in leukemia cells. Mitochondria are organelles that play a pivotal role in metabolic processes. Their dysfunction is related with many human disorders ranking from cancer to dementia [72,73], and the preservation of a viable pool of mitochondria is crucial to cell function. In addition, mitochondria are key mediators of tumorigenesis. For this reason, identifying new strategies able to mine mitochondrial cellular function may be a new attractive option to exploit in the cancer field. Recent studies revealed that p53 can influence mitochondrial function changing from normal to abnormal condition under different stress levels. Moreover, p53 has been described for its ability to translocate into mitochondria and to cause mitochondrial membrane depolarization and apoptosis. Accumulating evidences suggest that iron plays a critical role in mitochondrial functions, including energy metabolism and oxidative phosphorylation [74,75]. Mitochondria are sensitive to iron deprivation and the dynamic switch from fission to fusion is heavily dependent on this form of cellular stress [76]. Structural mitochondrial elements such as OPA1 and Drp1, responsible for mito-fusion or mito-fission respectively, regulate their activity according to iron level and they are directly involved, alongside p53 members, in the activation of the apoptotic signaling pathway [52,53,77]. The increased mitochondrial mass observed by confocal imaging showed a severe mitochondrial alteration in cells exposed to iron chelation, when compared to untreated cells, in line with the previously reported result [78]. The explanation for the mechanism could associated to an enhancement of the fusion process, through an inhibition of the fission process, or by a combination of both processes. Previous reports suggested that inhibition of mitochondrial fission significantly causes growth arrest and cell cycle inhibition, and the activation of p53 and p21 seems to be mainly responsible for this phenomenon [52,57,77]. Consistently, a previous study showed the ability of Drp1 to suppress p21 expression and to induce p53 degradation [57]. Finally, the Aco2 reduction observed in our experimental set could be an interesting pathway to be investigated in the field of leukemia. Indeed, Aco2 is the enzyme responsible for the isomerization of citrate to isocitrate in the Krebs Cycle and it binds one [4Fe-4S] cluster per subunit. Binding of a [3Fe-4S] cluster leads to an inactive enzyme. Aco2 activity in mitochondria is a sensitive redox sensor of reactive oxygen and nitrogen species in cells and its function is directly associated to iron levels [79]. Therefore, Aco2 could become a new interesting protein to investigate in order to identify a new vulnerable process to target in leukemia cells. On the other hand, the iron chelation seems to also cause an alteration in the mitochondria OxPhos activity, which appears lower and less efficient in comparison that in the untreated samples. This negative effect could depend directly on the chelation of iron that is a fundamental component of the cytochromes composing the electron transport chain. However, also, the disruption of the mitochondrial network could play a role in the mitochondrial dysfunction since it was demonstrated that isolated mitochondria are less efficient in terms of energy production with respect to those organized in a network [80,81].

All together, these results confirm the importance of aberrant mitochondrial metabolism activation to directly affect cancer cells. The identification of common mitochondrial targets for different types of AML could be attractive for new therapeutic approaches that could potentially be exploited, alone or in combination with other drugs, in patients with AML or MDS in order to improve responses or to delay progression.

## 4. Materials and Methods

### 4.1. Cell Culture Conditions

MOLM, NB4, and HL60 cell lines were purchased from American Type Culture Collection (ATCC, Manassas, VA, USA). MOLM and NB4 cells were grown in RPMI 1640 medium supplemented with 200 nmol/L Glutamine (EuroClone), 10% inactivated fetal bovine serum (FBS) (Sigma-Aldrich, St. Louis, MO, USA), and 0.1% penicillin/streptomycin. HL60 cells were grown in ISCOVE’s medium supplemented with 200 nmol/L Glutamine (EuroClone, Milan, Italy), 10% inactivated FBS, and 0.1% penicillin/streptomycin. All cell lines were maintained at 37 °C with 5% CO_2_. 

### 4.2. Patients Cohort

The local ethics committee of San Luigi Gonzaga (protocol 0003267, permission code 17/2016, 24 February 2016), in accordance with the Declaration of Helsinki, approved the study. After written informed consent, bone marrow (BM) and peripheral blood (PB) specimens were collected from 15 MDS patients, with a median age of 78 years (range 56–82), whose clinical and molecular features are summarized in Table 3. White blood cells (WBCs) were isolated by buffy coat. In vitro primary cultures were incubated with DFX (50 µM) for 48 h in complete ISCOVE’s medium and subsequently, apoptosis assay and *CDKN1A* and *PUMA* gene expression were evaluated as described below. p53 and p73 protein levels were evaluated by immunohistochemistry technique before and after DFX treatment in 5 additional patients, selected for the availability of samples before ICT and within one year of ICT, without any additional treatment including erythropoietic stimulating agents.

### 4.3. Cell Treatment and Calcein Fluorescence Assay

Cell lines and mononuclear cells isolated from patients at diagnosis were incubated with DFX (Selleckchem Italy, stock solution 50 mM/L in DMSO, Aurogene, Rome, Italy) for 48h at desired concentrations. After incubation, crucial experiments described below were performed. Labile iron pool (LIP) was quantified by exploiting its ability to bind to calcein acetoxymethyl ester (CA-AM). Notably, after 48 h of Deferasirox treatment, we incubated 1 × 10^6^ cells in PBS with calcein (Invitrogen), for 15 min at 37 °C. After three PBS washes, to remove excess reagent, we analyzed the samples by flow cytometry. LIP level was inversely proportional to measured fluorescence intensity. In normal condition, indeed, upon entering viable cells, calcein fluorescence is quenched by binding to cellular LIP. DFX acts by removing iron from its complex with CA. Therefore, the fluorescence emitted by the cells increases (measurable as Mean Fluorescence Intensity (MFI) with flow cytometry), thus confirming the occurred iron chelation.

### 4.4. Proliferation and Apoptosis Assay

Cell growth was evaluated by the MTT assay (Cell Proliferation Kit I (MTT), Sigma-Aldrich, St. Louis, MO, USA) [82], according to the manufacturer’s instructions. 50,000 cells/per well were seeded in triplicate for each condition, in a 96-well plate. 48 h after DFX treatment, MTT reagent was added inside each well and, after appropriate incubation, the corresponding absorbance was measured.

Apoptosis was evaluated by flow cytometry after labeling with fluorescein isothiocyanate (FITC)-conjugated annexin V and propidium iodide (Annexin V-FITC Apoptosis Detection Kit, Immunostep, Salamanca, Spain), as previously described [83]. BD CellQuest software (BD Biosciences) was used for data analysis of Annexin V-positive cells.

### 4.5. RNA Extraction and qRT-PCR Analysis

Total RNA was extracted using TRIzol Reagent (Ambion, Thermo Fisher Scientific, Waltham, MA, USA) as previously described [84]. Briefly, 1 µg of total RNA was used as template for the reverse transcription reaction. Expression levels of *CDKN1A*, *PUMA*, *GADD45*, and *MDM2* were evaluated with TaqMan technology (TaqMan Universal Master Mix, Thermo Fisher Scientific, Waltham, MA, USA), through the C1000 Thermal Cycler CFX96 Real-Time System (Bio-Rad, Hercules, CA, USA). qRT-PCR data were analyzed by Bio-Rad CFX Manager 3.1 software (Bio-Rad, Hercules, CA, USA). The analysis was performed in triplicate. Genes’ expression was normalized with respect to the *ABL* housekeeping gene and expressed as 2^–ΔΔCt^. Universal human references RNA (Stratagene, San Diego, CA, USA) was used to calibrate the assay.

### 4.6. MitoTracker Staining and Morphological Analysis of Mitochondria

MitoTracker Green FM (M7514) was dissolved in DMSO to obtain 1 mM stock solutions. To stain our cell lines, MitoTracker solution was added to the growth medium, after 48 h of incubation with DFX at 1:5000 dilution (to a final working concentration of 100 nM). After 40 min of incubation, cells were centrifuged and resuspended in fresh prewarmed medium, to be analyzed by confocal scanning microscope (LSM 5110; Carl Zeiss MicroImaging Inc., Oberkochen, Germany, 63× objective) [54]. The morphological analysis of mitochondria was performed in silico using existing ImageJ plug-ins following Valente et al.’s toolset, called MiNA (Mitochondrial Network Analysis). MiNA allows semi-automated analysis and consists in images’ preprocessing, to ensure quality, conversion to binary image, and in the production of the final skeleton for the quantitative analysis. Briefly, images were opened on ImageJ and processed as follows: 1-Process/Filters/Unsharp Mask; 2-Process/Enhance Local Contrast (CLAHE); 3-Process/Filters/Median; 4-Process/Binary/Make Binary; 5-Process/Binary/Skeletonize; 6-Analyze/Skeleton/Analyze Skeleton (2D/3D); 7-Plugins/StuartLab/MiNA Scripts/MiNA Analyze Morphology.

### 4.7. Immunofluorescence Assay

After cytospin, 50,000 cells were fixed in 4% PFA for 10 min [85]. Cells were permeabilized with 0.5% triton for 5 min, blocked for 45 min with PBS 10% BSA and subsequently incubated for 2 h with the specific primary antibodies (p53 DO-1 (sc-126, Santa Cruz Biotechnology, Dallas, TX, USA) and p73 (PA5-35368, ThermoFisher Scientific, Waltham, MA, USA). Immunocomplex was detected by 40 min incubation with secondary antibody (Alexa Fluor 488 or 543, Invitrogen). PI was used for nuclear staining. Cells were visualized with a confocal scanning microscope (LSM 5110; Carl Zeiss MicroImaging Inc., Oberkochen, Germany, 63× objective) and pictures were quantified by the Java (Image J) program.

### 4.8. Protein Extraction and Immunoblotting

To isolate total protein content, samples were lysed on ice with RIPA buffer (50 mmol/L Tris-HCl pH 8.0, 150 mmol/L NaCl, 1% Np40, 0.5% DOC, 0.1% SDS, freshly added to protease and phosphatase inhibitors cocktail). Cell debris were removed by centrifugation at 14,000× *g* at 4 °C for 15 min. Protein concentration was determined by Bio-Rad Protein Assay Bio-Rad, Hercules, CA, USA) [86]. Fifty µg of each total cell lysate were loaded, resolved through SDS-PAGE 8% or 12% gel, and electroblotted onto 0.2 µm nitrocellulose membranes (Bio-Rad, Hercules, CA, USA). After blocking with 5% BSA (Sigma-Aldrich, St. Louis, MO, USA) in TBS (Tris-HCl pH 7.4, 150 mM NaCl), 0.3% Tween-20 for at least 1 h at RT, membranes were incubated overnight (ON) at 4 °C with the primary antibodies (Aconitase 2, #6922, Cell Signaling Technology, Danvers, MA, USA, p21 sc-6246, Santa Cruz Biotechnology, Dallas, TX, USA, MDM2, sc-13161, Santa Cruz Biotechnology, Dallas, TX, USA, Cleaved Caspase 3, #9664, Cell Signaling Technology, Danvers, MA, USA, GAPDH, sc-365062, Santa Cruz Biotechnology, Dallas, TX, USA). For each antibody, a dilution of 1:1000 was used. As secondary antibodies, we used peroxidase-conjugated goat anti-mouse IgG-HRP (Santa Cruz Biotechnology, sc-2005) or goat anti-rabbit IgG-HRP (Santa Cruz Biotechnology, sc2004), both at 1:8000 dilution for 1 h at RT. Immuno-reactive bands were visualized by using chemiluminescent enhanced reagent (Clarity Western ECL Substrate #170-5061, Bio-Rad). Quantification was performed using the Image Lab program (BioRad Laboratories, Hercules, CA, USA) [87].

### 4.9. Gene Expression Analysis in Deferasirox-Treated Cells

Gene expression profiling of DFX-treated leukemic cells by Affymerix GeneChip (U133 Plus 2.0, Santa Clara, CA, USA) was retrieved from GEO (GSE11670). Treated samples (GSM296615, GSM296616) with Deferasirox 50 µM for 16 h and untreated controls (GSM296608, GSM296609) were analyzed by the GEO2R tool. Original submitter-supplied processed data tables were imputed into R using GEO query [88]. Linear Models for Microarray Analysis (Limma) R package from the Bioconductor project were used to compute differential expression [89]. Differentially expressed probes (*p*-value < 0.001) were annotated with the Ensembl BioMart tool and the GRCh38.p12 as a reference genome assembly [90]. In case of multiple probe sets corresponding to the same gene annotation, the probe set with lower *p*-value was used. Identified differentially expressed genes were submitted to a pathway analysis using the tool EnrichR [91]. The first 10 enriched terms from the National Cancer Institute Nature (NCI-Nature) Pathway Interaction Database (PID) 2016 were ordered by ascending adjusted *p*-value and reported [92]. Differentially expressed genes involved in the identified pathways were visualized and hierarchically clustered by using the Clustergrammer web-tool integrated in EnrichR [93].

### 4.10. Evaluation of ATP/AMP Ratio as Marker of Cellular Energy Status

To evaluate the cellular energy status, the ATP/AMP ratio was calculated on the basis of the ATP and AMP intracellular level. ATP and AMP concentrations were measured spectrophotometrically at 340 nm, following the NADP reduction or NADH oxidation, respectively [75].

### 4.11. Oxygen Consumption Rate (OCR), ATP Synthesis, and P/O Ratio Evaluation

The oxygen consumption rate (OCR) was measured in a closed chamber, using an amperometric electrode (Unisense-Microrespiration, Unisense A/S, Tueager, Denmark). For each experiment, 100,000 cells, treated or not with 50 μM Deferasirox for 48 h, were permeabilized with 0.03% digitonin for 1 min. 5 mM pyruvate and 2.5 mM malate were used to stimulate the pathway composed by complexes I, III, and IV, while 20 mM succinate induced the pathway formed by complexes II, III, and IV. To observe the ADP-stimulated respiration rates, 0.08 mM ADP was added after pyruvate and malate or succinate addition [75]. ATP synthesis, through F0-F1 ATP synthase, was investigated by the luciferin/luciferase chemiluminescent method (luciferin/luciferase ATP bioluminescence assay kit CLSII, Roche, Basel, Switzerland), with ATP standard solutions between 10^−8^ and 10^−5^ M. The reaction was monitored in a luminometer (GloMax^®^ 20/20n Luminometer, Promega Italia, Milano, Italy) [75]. The OxPhos efficiency (P/O ratio) was calculated as the ratio between the concentration of the produced ATP and the amount of consumed oxygen in the presence of respiratory substrate and ADP. When the oxygen consumption is completely devoted to the energy production, the P/O ratio should be around 2.5 and 1.5 after pyruvate + malate or succinate addition, respectively.

### 4.12. Immunohistochemistry on MDS Bone Marrow Samples

Immunohistochemistry experiments were performed on formalin-fixed [94], paraffin-embedded serial sections of 4 μm thick, derived from 5 MDS patients at diagnosis and within one year of iron chelation treatment (ICT). FFPE tissue sections were previously deparaffinized. p73 (dilution 1:100, ab40658) and p53 (dilution 1:200, MA5-12453) labelling was performed using the UltraView Universal DAB Detection Kit (Ventana Medical Systems, Orovalley, AZ, USA) on Benchmark ULTRA (Roche, Ventana, Meylan, France). Slides were scored independently by 2 pathologists.

### 4.13. Statistical analyses

Statistical analyses were performed using the paired *t*-test. All the experiments were performed in triplicate and analyses with confidence level greater than 95% are indicated as significant and marked as follows: * *p* ≤ 0.05, ** *p* ≤ 0.01, and *** *p* ≤ 0.001. Biochemical data were analyzed by one-way analysis of variance (ANOVA) followed by Tukey’s multiple comparison test, using GraphPad Prism version 7.00. *p*-value < 0.05 was considered significant.

## 5. Conclusions

In conclusion, based on our findings, we proposed a new model in which iron chelation, besides reducing oxidative stress level, could play a key role in p53 and p73 stability, via MDM2 reduction. Although, even if they are not conclusive and may need to be confirmed by further studies, primarily in order to better clarify the potential role of MDM2 in the present context, the in vitro anti-leukemic effect of Deferasirox, accompanied by a significant increase in p53 family target genes, seemed very interesting. All these events may therefore exert antitumor effects and could explain the potential benefits of ICT in improving OS and delaying leukemic transformation in MDS and AML patients.

## Figures and Tables

**Figure 1 ijms-21-07674-f001:**
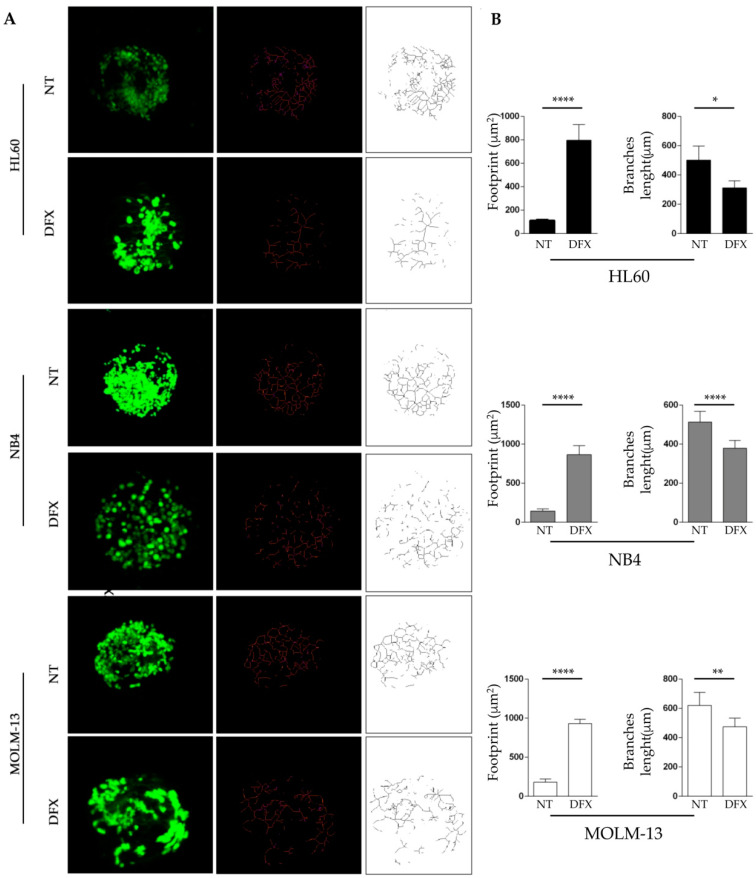
Iron chelation induces an altered network in acute myeloid leukemia cell lines. HL60, NB4, and MOLM-13 were treated for 48 h with 50 µM of DFX and subsequently incubated with 100 nM of MitoTracker green for mitochondrial network analysis. (**A**) The original green image obtained with confocal microscope is a three-dimensional (3D) image created with z-stack project (63× magnification). It has been processed using the MiNA toolset to generate an accurate skeleton. In red and black and white, the skeleton of mitochondria of the three cell lines treated with DFX is visible, compared to respective control. (**B**) Quantification of mitochondria footprint and branches length, which corresponds respectively to the area and connection, of mitochondria expressed in µm. Abbreviations: NT, not treated; DFX, Deferasirox. * *p* ≤ 0.05, ** *p* ≤ 0.01, and **** *p* ≤ 0.0001.

**Figure 2 ijms-21-07674-f002:**
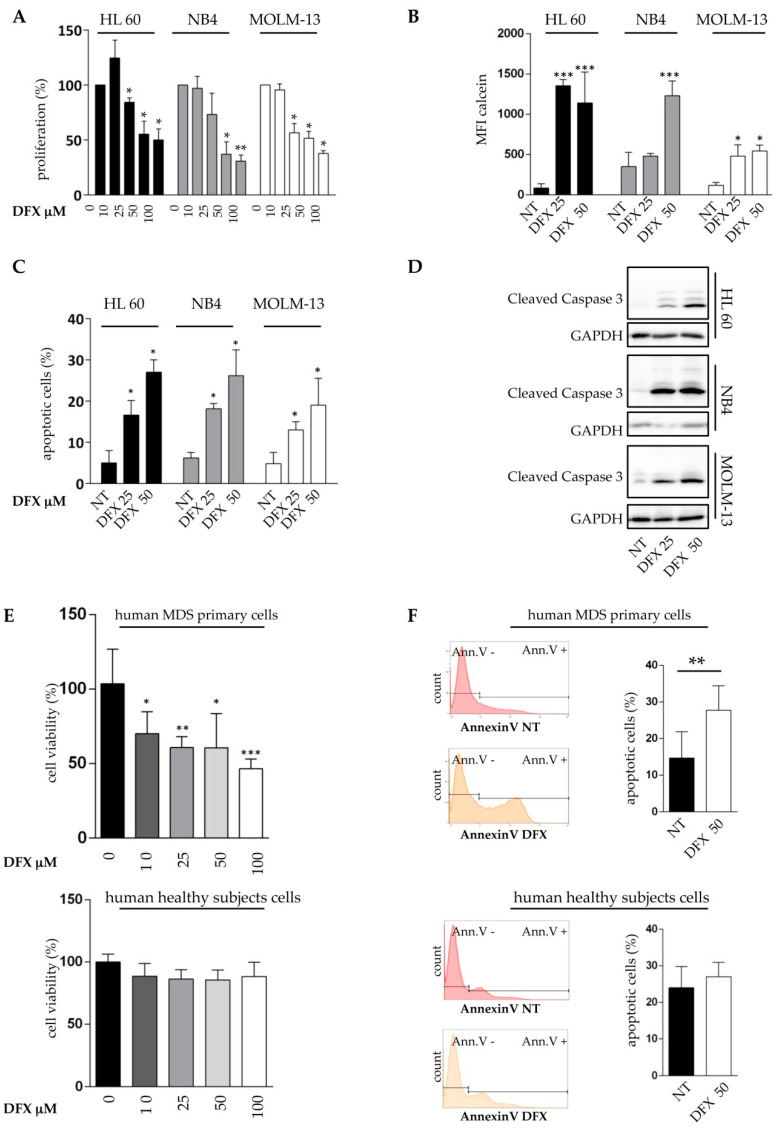
Deferasirox exerts an anti-leukemic activity on AML cell lines and on MDS patients’ cells. (**A**) HL60, NB4, and MOLM-13 were treated with 0, 10, 25, 50, or 100 μM DFX for 48 h and the MTT assay was performed to evaluate the proliferation index. The percentage of proliferation is expressed after normalizing with untreated cells (100%). (**B**) Mean fluorescence intensity (MFI) of FITC-calcein signal obtained after 48 h of DFX treatment. LIP level was inversely proportional to measured fluorescence intensity. (**C**) Percentage of apoptosis evaluated by flow cytometry after FITC Annexin-V assay on HL60, NB4, and MOLM-13 treated with 25 and 50 μM of DFX. (**D**) Western blot analysis of cleaved caspase 3 revealed that its amount increased in a DFX dose-dependent fashion, in HL60, NB4, and MOLM-13. (**E**) Five MDS/AML patients or 5 healthy subjects were treated with 10, 25, 50, or 100 μM of DFX for 48 h and the MTT assay was performed. (**F**) Representative histograms and % of apoptosis evaluated by flow cytometry after FITC Annexin-V assay on 5 MDS/AML patients or 5 healthy subjects treated with 50 μM of DFX. Abbreviations: NT, not treated; DFX, Deferasirox; Ann V, Annexin V. * *p* ≤ 0.05, ** *p* ≤ 0.01, and *** *p* ≤ 0.001.

**Figure 3 ijms-21-07674-f003:**
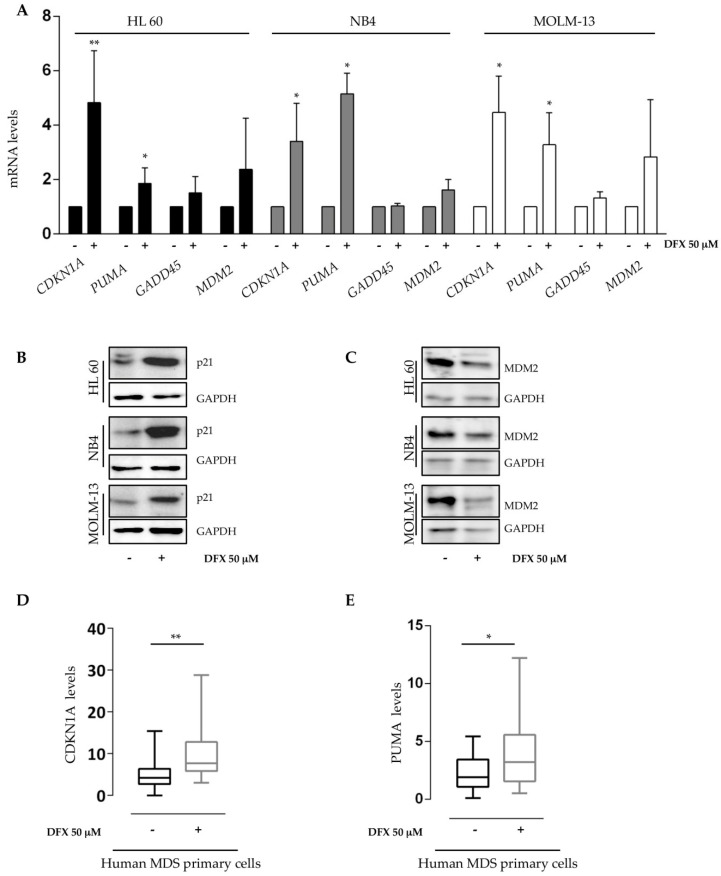
Deferasirox activates p53 targets on acute myeloid leukemia cell lines and MDS/AML patients. (**A**) *CDKN1A*, *PUMA*, *GADD45*, and *MDM2* gene expression were assayed by qRT-PCR in HL60, NB4, and MOLM-13 after 48 h treatment with DFX 50 μM. The amount is expressed as fold changes compared to untreated cells after normalizing on the ABL housekeeping gene. (**B**,**C**) Western blot analysis of target proteins p21 and MDM2 after iron chelation treatment confirmed the effects of DFX on the p53 pathway. (**D**,**E**) *CDKN1A* and *PUMA* gene expression was assayed by qRT-PCR on 15 patients’ cells after 48 h in vitro treatment with DFX 50 μM. The mRNA quantity is expressed as 2^−ΔΔCt^ after normalization with the ABL housekeeping gene. Abbreviations: -, not treated; + treated with DFX 50 μM; DFX, Deferasirox. * *p* ≤ 0.05, ** *p* ≤ 0.01.

**Figure 4 ijms-21-07674-f004:**
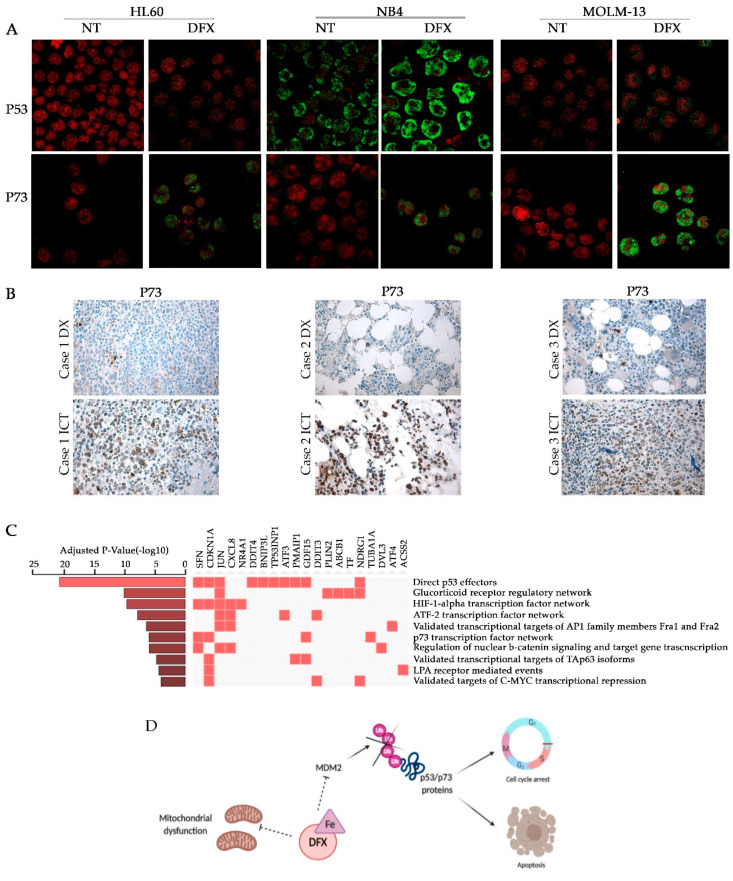
Deferasirox regulates p53 and p73 protein stability. (**A**) Immunofluorescence of p53 and p73 after DFX treatment. The green signal corresponds to p53 or p73 while the red propidium is used to detect nuclei (63× magnification). (**B**) p73 Immunohistochemistry on MDS bone marrow samples derived from 3 different patients at diagnosis (DX) and within one year of iron chelation treatment (ICT) (20× magnification). (**C**) Differentially expressed genes clusters enriched according to Gene Ontology terms and ordered by ascending adjusted *p*-value. (**D**) Schematic representation of the plausible role of iron chelation on mitochondrial activity and on p53 family stability (created in biorender.com). Abbreviations: NT, not treated; DFX, Deferasirox.

**Table 1 ijms-21-07674-t001:** Iron chelation induced a decrement of oxygen consumption, ATP synthesis and a less efficient OxPhos in acute myeloid leukemia cell lines. The table reports the oxygen consumption rate (OCR) and the ATP synthesis (ATPsynth) through F0-F1 ATP synthase in acute myeloid leukemia cell lines untreated (NT) or treated with DFX. These activities have been evaluated in the presence of pyruvate/malate (P/M) or succinate (Succ), to investigate the OxPhos pathways triggered by complex I or complex II, respectively. The P/O value is calculated as the ratio between ATPsynth and OCR and represents a marker of mitochondrial efficiency. Literature reports that a complete efficiency is observed when the P/O ratios are 2.5 or 1.5 in the presence of pyruvate/malate or succinate, respectively [55]. Data are expressed as mean + standard deviation (SD) and are representative of three independent experiments. *** or **** indicate a *p* < 0.001 or *p* < 0.0001 respectively, between the same untreated or DFX-treated samples.

	HL60		NB4		MOLM-13	
	NT	DFX	NT	DFX	NT	DFX
**OCR_P/M (nmol O/min/10^6^ cells)**	33.86 ± 1.25	19.51 ± 1.84 ****	30.98 ± 1.74	14.03 ± 1.23 ****	18.55 ± 1.19	9.96 ± 0.89 ***
**ATPsynth_P/M (nmol ATP/min/10^6^ cells)**	82.89 ± 0.91	37.46 ± 1.86 ****	75.98 ± 3.11	16.58 ± 1.84 ****	44.96 ± 1.77	11.24 ± 0.93 ***
**P/O_P/M**	2.45 ± 0.08	1.62 ± 0.03 ****	2.45 ± 0.12	1.18 ± 0.04 ****	2.43 ± 0.08	1.13 ± 0.04 ***
**OCR_Succ (nmol O/min/10^6^ cells)**	22.50 ± 1.08	12.87 ± 0.73 ****	20.52 ± 1.02	9.26 ± 0.85 ****	12.58 ± 0.54	6.78 ± 0.47 ****
**ATPsynth_Succ (nmol ATP/min/10^6^ cells)**	34.96 ± 2.85	14.95 ± 0.58 ****	32.05 ± 0.68	6.99 ± 0.31 ****	18.97 ± 1.45	4.73 ±0.66 ****
**P/O_Succ**	1.56 ± 0.09	1.00 ± 0.05 ****	1.57 ± 0.07	0.76 ± 0.04 ****	1.55 ± 0.09	0.72 ± 0.04 ****

**Table 2 ijms-21-07674-t002:** Iron chelation induced a decrement of cellular energy status in acute myeloid leukemia cell lines. The table reports the intracellular level of ATP and AMP, and the consequent ATP/AMP ratio to investigate the cellular energy status. Data are expressed as mean + SD and are representative of three independent experiments. *** or **** indicate a *p* < 0.001 or *p* < 0.0001 respectively, between the same untreated or DFX-treated samples.

	HL60		NB4		MOLM-13	
	NT	DFX	NT	DFX	NT	DFX
**ATP (mM/mg)**	2.39 ± 0.09	1.58 ± 0.22 ***	2.24 ± 0.10	1.29 ± 0.04 ***	2.17 ± 0.06	1.23 ± 0.04 ***
**AMP (mM/mg)**	0.84 ± 0.03	1.21 ± 0.04 ***	0.83 ± 0.04	1.40 ± 0.06 ***	0.95 ± 0.05	1.47 ± 0.02 ***
**ATP/AMP**	2.86 ± 0.07	1.45 ± 0.18 ****	2.71 ± 0.21	0.92 ± 0.06 ****	2.28 ± 0.16	0.84 ± 0.03 ****

**Table 3 ijms-21-07674-t003:** Clinical and molecular features of patients and healthy donors enrolled in the study.

Age, Years Median 78 (Range 56–82)	Diagnosis	Karyotype	Bm Blast (%)	Ferritin (ng/mL)
81	MDS-SLD	Normal	2.0	N/A
58	Isolated del (5q)	46, XX, 5q-	1.5	1191
67	AML	Normal	25.0	2250
78	MDS-MLD	46, XY, del 9 (q22:q32)	2.2	2292
70	MDS-RS-SLD	47, XY, +8	2.0	2587
82	MDS-EB-II	Normal	5.0	975
69	MDS-EB-II	N/A	15.0	700
56	MDS-MLD	N/A	3.0	4706
78	MDS-EB-I	Normal	7.0	1207
69	MDS-SLD	N/A	N/A	2643
64	MDS-MLD	Normal	4.0	1314
71	MDS-SLD	Normal	3.0	N/A
79	Isolated del (5q)	46, XY, 5q-	2.5	746
82	MDS-EB-I	Normal	N/A	N/A.
81	AML	N/A	20.0	2272
63	Healthy donor 1	Normal	-	200
52	Healthy donor 2	Normal	-	13
56	Healthy donor 3	Normal	-	80
57	Healthy donor 4	Normal	-	83
53	Healthy donor 5	Normal	-	49

Abbreviation: MDS-SLD, MDS with single lineage dysplasia; AML, acute myeloid leukemia; MDS-RS-SLD, MDS with ring sideroblasts with single lineage dysplasia; MDS-MLD: MDS with multilineage dysplasia; MDS EB I: MDS with excess of blast type me; MDS EB II: MDS with excess of blast type II; N/A: not available.

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
