# Peer review of "Deferasirox-Dependent Iron Chelation Enhances Mitochondrial Dysfunction and Restores p53 Signaling by Stabilization of p53 Family Members in Leukemic Cells"

_ijms, 2020, doi:10.3390/ijms21207674_

Round 1

Reviewer 1 Report

Nic pre-clinical study on cell lines and primary cell cultures exploring why iron-chelation works. Probably role of MDM2 should be explored formally in subsequent studies (MDM2 null cell lines, siRNA targeting MDM2, a small molecule inhibiting MDM2 etc) as only an asssociation is presented based for MDM2 protein expressions rather than cause-effect relationship. (by the way the lack of cause-effect relationship is also apperatent for p73/p63). However, I believe that the data is enough for a publication for its implication of p53 family molecules for the effects of iron-chelation. Minor comment: One of the cell line is an AML (sample 3)L, it may be a good idea to correct tittle 3.2 in results section mentioning only MDS

Author Response

Response to Reviewer 1 Comments

We appreciate the positive and valuable feedback from the reviewer.

 Point 1: nic pre-clinical study on cell lines and primary cell cultures exploring why iron-chelation works. Probably role of MDM2 should be explored formally in subsequent studies (MDM2 null cell lines, siRNA targeting MDM2, a small molecule inhibiting MDM2 etc) as only an asssociation is presented based for MDM2 protein expressions rather than cause-effect relationship. (by the way the lack of cause-effect relationship is also apperatent for p73/p63). However, I believe that the data is enough for a publication for its implication of p53 family molecules for the effects of iron-chelation.

Response 1: we really thank the reviewer for the comments and we are very pleased that on the whole the manuscript has been appreciated. To emphasize the putative role of MDM2 we added a sentence in the conclusion section, in order to indicate our intention to explore more intensively in future the MDM2-p53-iron relationship.

Minor comment:

Point 2: one of the cell line is an AML (sample 3), it may be a good idea to correct title 3.2 in the results section mentioning only MDS.

Response 2: regarding this suggestion, the AML sample 3 is indicated in table I including the list of patients enrolled in this study, and not to cell lines. Therefore, as you noted, patients are almost entirely MDS while cell lines are AML. For this reason, we consider to maintain the title 3.2: “Deferasirox exerts in vitro anti-leukemic activity on acute myeloid leukemia cell lines and on primary MDS cells “but we change the title of Figure 2 as follows: “Deferasirox exerts an anti-leukemic activity on AML cell lines and on MDS cells”.

Reviewer 2 Report

Summary: In the present study, the authors sought to understand the mechanism by which the oral iron chelator, Deferasirox, improves overall survival and delayed leukemia transformation in patients with myelodysplastic syndrome (MDS). Using qRT-PCR, they show that p53 target genes are upregulated when AML or MDS cells are treated with Deferasirox, and this correlates with changes in the mitochondrial network.

Major Comments:

  • The Introduction in its current form is difficult to read. Revision is recommended. What is the problem? What are the knowns? What are the unknowns? What have you done about it?
  • A clear hypothesis should be stated somewhere in the Introduction.
  • Does Deferasirox have any effect on survival or apoptosis of normal stem and progenitor cells? What is the therapeutic window?
  • Does Deferasirox have any effect on the K562 CML cell line. TP53 is mutationally inactivated in K562 cells. Therefore, if Deferasirox is dependent on p53, it should have no effect.
  • In lines 314-315, the authors state, “Within this scenario, cell cycle inhibition and induction of apoptosis seemed to be the events most affected, directly dependent on p53 family activity.” I don’t see data to substantiate this claim.
  • There is no clear evidence that p53, p63, or p73 are responsible for the effects of Deferasirox in AML and MDS cells.
  • Lines 28-30 of the abstract reads, “Our findings have indicated that DFX can lead to apoptosis, to impairment of cell growth, and can induce a significant alteration of mitochondria network, with a sharp reduction in mitochondrial activity.” Where is data demonstrating a sharp reduction in mitochondrial activity?

Minor Comments:

  • Phosphorylation is misspelled on line 409.

Author Response

Response to Reviewer 2 Comments

We appreciate the positive and valuable feedback from the reviewer.

Point 1: the Introduction in its current form is difficult to read. Revision is recommended. What is the problem? What are the knowns? What are the unknowns? What have you done about it? A clear hypothesis should be stated somewhere in the Introduction.

Response 1: we genuinely and truly thank the referee for this helpful comment. To properly address this issue, in the revised version the section Introduction has been thoroughly and extensively revised. Furthermore by keeping into account the referee request we follow the lineup suggested.

Point 2: does Deferasirox have any effect on survival or apoptosis of normal stem and progenitor cells? What is the therapeutic window?

Response 2: we frankly thank the reviewer for the comment. From a metabolic point of view, the normal stem and progenitor cells are characterized by anaerobic metabolism and low metabolic rate due to the hypoxic environment of the bone marrow niche (Suda et al. 2011; Zhang and Sadek 2014). Moreover, in general, the undifferentiated cells principally perform the anaerobic glycolysis to avoid the oxidative damage associated with the mitochondrial aerobic function (Ravera et al. 2018; Simsek et al. 2010; Ahlqvist et al. 2015). Therefore, since the iron chelation exerts its negative effect on the aerobic metabolism, it is possible to argue that Deferasirox treatment may not affect the stem and progenitor cells, at least as long as they remain in the bone marrow niche.

Point 3: does Deferasirox have any effect on the K562 CML cell line. TP53 is mutationally inactivated in K562 cells. Therefore, if Deferasirox is dependent on p53, it should have no effect.

Response 3: we thank the referee for this observation. In our paper, we decide to use NB4 and HL60 because they are mutated and deleted for p53 respectively. To fulfill the reviewer request we performed proliferation and apoptosis assays on K562, and we are going to confidentially enclose the results below. In K562 Deferasiox is able to induce a significant reduction in proliferation and an increase in apoptosis. Moreover, p73 protein signal significantly increased after treatment. These results confirmed the possibility that even in K562 alternative pathways, like p73 reactivation, can overtake p53 activity, corroborating our observations.

Point 4: in lines 314-315, the authors state, “Within this scenario, cell cycle inhibition and induction of apoptosis seemed to be the events most affected, directly dependent on p53 family activity.” I don’t see data to substantiate this claim.

Response 4: regarding this suggestion, we modified the sentence, as follows (line 693-694): “Thus, it is plausible hypnotizer a real implication of p53 family, even if we don’t exclude the existence of other mechanisms involved”, hoping to meet the reviewer request.

Point 5: there is no clear evidence that p53, p63, or p73 are responsible for the effects of Deferasirox in AML and MDS cells.

Response 5: we thank the reviewer for this observation. In this regard, we implemented figure 4 with immunohistochemistry assays performed against p73 and p53 on 5 MDS patients at diagnosis and within one year of treatment with Deferasirox (Figure 4B and Supplementary Figure S2 A). In addition, we added a table with the corresponding scores (Supplementary Figure S2 B). Consistently, even after DFX treatment, we observed a significant increase in p73 protein level. In our specimens, p53 is extremely difficult to detect, unless that mutated. Thus the figure has only p73 panel. We have assumed that being extremely crucial to several cellular processes, therefore even small changes in its amount are enough to activate a response. However, in Supplementary figure S2 C we also included some patients' immunofluorescences on p53 and p73 after 48 hours of in vitro treatment.  

Point 6: lines 28-30 of the abstract reads, “Our findings have indicated that DFX can lead to apoptosis, to impairment of cell growth, and can induce a significant alteration of mitochondria network, with a sharp reduction in mitochondrial activity.” Where is data demonstrating a sharp reduction in mitochondrial activity?

Response 6: In the revised version, we added two tables that report the oxygen consumption rate (OCR), ATP synthesis through F0-F1 ATP synthase, and the consequent P/O value, as a marker of mitochondrial activity efficiency. These activities have been evaluated in the presence of pyruvate + malate or succinate to investigate, respectively, the oxidative phosphorylation activity triggered by respiratory complex I or complex II. Moreover, we evaluated the intracellular concentration of ATP and AMP, calculating the ATP/AMP ratio as a marker of cellular energy status. Data demonstrate that the DFX treatment induced a decrement of OCR and ATP synthesis in all cell lines. Moreover, the P/O ratio after DFX treatment appeared lower in comparison to that of untreated samples, suggesting a minor efficiency of the OxPhos activity. This energy metabolism alteration also determined a decrement of the ATP/AMP ratio due to the low ATP synthesis and the increment of AMP level.

Minor Comments:

Point 1 and Response 1: line 409: "Phosphorilation” was replaced with " Phosphorylation ".

Reviewer 3 Report

Iron homeostasis is important for many cells independent of origin, and the disruption of this balance commonly results in reduced cellular proliferation and cell death. Along this line, the detrimental effect of iron chelation on cell proliferation is an issue largely explored. Despite of numerous results, many of the conclusions of this work are predictable and/or previously reported. More in detail, the disruption of mitochondrial functionality, the impairment of AML cell lines growth (Callens C et al 2010) and p53 protein stabilization as consequence of iron chelation were previously shown by other authors.

Some of the findings described in the manuscript are still potentially interesting because they are novel, such as the involvement of p73, or because they are shown in AML cells for the first time. However, the study would further benefit from additional experiments:

Mitochondrial disfunction induced by iron starvation should be better addressed, showing functional characterization. I also suggest to enrich the pool of enzyme activities analyzed. Succinate dehydrogenase might be good candidate. See comment below.

In figure 2, authors refer their experiment as aimed at measuring cell proliferation. However, after 48h growth they plated equal number of cells in MTT assay. In this way, authors indirectly measure overall cellular dehydrogenase activity rather than cell number. The experiment should be done by plating equal volume from cell culture. How did they perform experiments illustrated in figure S1B? It’s not clear from the methods. In my files, the legend to figure S1 is missing.

The same is true for patients cells. Do primary cells proliferate in vitro? Moreover, in the experiments made on patients cells, the comparison with control cells should be shown. Do patients cells have higher iron content than WBC isolated from healthy subjects in basal growth condition? Authors should clarify this point to increase the relevance of their work.

LIP measurement was not performed using the full canonical technique. While it is true that higher fluorescence in deferasirox treated cells means lower LIP levels, a final step in the assay must be performed by adding a lipophilic iron chelator. This, will also be useful for comparing baseline LIP levels between patients and control cells.

Figure 3. Why do they present mRNA levels of MDM2 in supplemental? The contrast between the increase of mRNA levels (predicted as compensatory mechanism of p53 stabilization) and the reduction of protein level is an important point to mechanistically explain the effect of iron chelation worth to be further investigated. Does iron chelation induce increased MDM2 protein degradation?

Figure 4. The sentence “deferasirox increased both p53 and p73 levels” (Line 345) is ambiguous. It makes figure 4A difficult to understand since deferasirox does not increase p53 in p53 null cell lines. Furthermore, immunofluorescence does not clearly show an increase in proteins but only nuclear localization. A WB should be done at least for p73 to asses whether deferasirox increases protein levels.

Minor points

Data presentation in Figure S2C is more explicative than that of figure 2B

Which cell line was used for gene expression data sets?

Line 363, 3C should be 4C.

In REF 27, year of publication and DOI are missing

Author Response

Response to Reviewer 3 Comments

Point 1: mitochondrial disfunction induced by iron starvation should be better addressed, showing functional characterization. I also suggest to enrich the pool of enzyme activities analyzed. Succinate dehydrogenase might be good candidate. See comment below

Response 1: we truly thank the referee and we apologize for the missed data. In the revised manuscript to better characterize the mitochondrial dysfunction, we added two tables that report the oxygen consumption rate (OCR), ATP synthesis through F0-F1 ATP synthase, and the consequent P/O value, as a marker of mitochondrial activity efficiency. Moreover, we evaluated the intracellular concentration of ATP and AMP, calculating the ATP/AMP ratio as a marker of cellular energy status. Data demonstrate that the DFX treatment induced a decrement of OCR and ATP synthesis in all cell lines. Moreover, the P/O ratio after DFX treatment appeared lower in comparison to that of untreated samples, suggesting a minor efficiency of the OxPhos activity. This energy metabolism alteration also determined a decrement of the ATP/AMP ratio due to the low ATP synthesis and the increment of AMP level.

Point 2: in figure 2, authors refer their experiment as aimed at measuring cell proliferation. However, after 48h growth they plated equal number of cells in MTT assay. In this way, authors indirectly measure overall cellular dehydrogenase activity rather than cell number. The experiment should be done by plating equal volume from cell culture. How did they perform experiments illustrated in figure S1B? It’s not clear from the methods. In my files, the legend to figure S1 is missing

Response 2: we are very thankful to the reviewer for this comment. We modified the “2.4 Proliferation and apoptosis assay” section to clarify the procedure followed. Briefly, MTT assay is a method used for measuring cell-proliferation, viability, and cytotoxicity. The color formation, measured by reading the optical density of the sample at a wavelength of 590 nm is proportional to the NADPH-dependent reductase cellular activity, which in turn is proportional to the number of proliferating and viable cells. In our experiment, we started by seeding at time 0 the same number of cells/well (50,000) for each condition (Not treated and Deferasirox at different concentrations). 48 hours later, after Deferasirox treatment, MTT was measured, within the same well. In this way, if Deferasirox impaired the number of cells, in terms of reduction of proliferation and/or reduction of viable cells, we will obtain a decrease in the absorbance value. The result reported in figure Supplementary S1B represents the count of viable cells performed by FACS, by discriminating the viable by dead cells according to their morphology and size characteristic. Therefore, these findings may overlap with the proliferation assay.

Point 3: the same is true for patients cells. Do primary cells proliferate in vitro? Moreover, in the experiments made on patients cells, the comparison with control cells should be shown. Do patients cells have higher iron content than WBC isolated from healthy subjects in basal growth condition?

Response 3: we really thank the reviewer for the comments. For primary human cells is extremely difficult to proliferate in vitro in 48 hours since their proliferation rate is negligible in 48 hours. Indeed, it would be better to consider that it this case MTT is measuring the number of viable cells and not their proliferation rate. Therefore, we also modified the y-axis title.

In this regard, we implemented Figure 2 (panel E and F) and the result section (line 634-644) by including proliferation and apoptosis assays performed on 5 healthy subjects. We added normal samples enrolled in Table 1, where we indicated their ferritin levels (a standard indirect parameter of WBC iron content) at the time of the assays. The levels are totally different from those of MDS/AML patients samples, confirming that the iron content of WBC isolated from healthy subjects is extremely low. Thus, our results showed that DFX is not able to induce apoptosis or anti-proliferative effects in healthy cells. Thus, iron overload and p53 family inactivation could be a potential peculiarity of MDS/AML leukemic cells, representing a new target to exploit by iron chelation.

Point 4: LIP measurement was not performed using the full canonical technique. While it is true that higher fluorescence in deferasirox treated cells means lower LIP levels, a final step in the assay must be performed by adding a lipophilic iron chelator. This, will also be useful for comparing baseline LIP levels between patients and control cells.

Response 4: we are very thankful to the reviewer for this comment. We are aware that the canonical technique of LIP measurement includes the use of a lipophilic iron chelator to have an absolute measure of LIP inside the sample. However, our aim was to assess a relative value of LIP, in order to validate the drug efficacy.

Point 5: figure 3. Why do they present mRNA levels of MDM2 in supplemental? The contrast between the increase of mRNA levels (predicted as compensatory mechanism of p53 stabilization) and the reduction of protein level is an important point to mechanistically explain the effect of iron chelation worth to be further investigated. Does iron chelation induce increased MDM2 protein degradation?                                                                                                                        Response 5: We thank the reviewer for that relevant comment. We added in figure 3A, besides CDKN1A, PUMA and GADD45 expression, MDM2 expression, in order to emphasize better the relevance of the p53-MDM2 negative feedback loop. 

Point 6: figure 4. The sentence “Deferasirox increased both p53 and p73 levels” (Line 345) is ambiguous. It makes figure 4A difficult to understand since deferasirox does not increase p53 in p53 null cell lines. Furthermore, immunofluorescence does not clearly show an increase in proteins but only nuclear localization. A WB should be done at least for p73 to assess whether deferasirox increases protein levels.                                          

Response 6: according to the reviewer's suggestion, we modified this section (line 743-747). In addition, we added in Supplementary Figure S1 p73 western blot. (Figure S1 E)

Minor points

Point 1: data presentation in Figure S2C is more explicative than that of figure 2B.

Response 1: To satisfy the reviewer request we substitute figure 2B with S2C. Consequently, 2B has been inserted in supplementary.

Point 2: which cell line was used for gene expression data sets?

Response 2: The cell line used is K562. To better clarify this point we added inside the text the name of the cell line used (line 756)

Point 3: line 363: 3C was substituted with 4C

Point 4: ref 27: the year of publication and DOI have been added

Round 2

Reviewer 2 Report

Summary: In the present study, the authors sought to understand the mechanism by which the oral iron chelator, Deferasirox, improves overall survival and delayed leukemia transformation in patients with myelodysplastic syndrome (MDS). Using qRT-PCR, they show that p53 target genes are upregulated when AML or MDS cells are treated with Deferasirox, and this correlates with changes in the mitochondrial network.

Major Comments:

  • The 3rd paragraph of the introduction is still very difficult to follow, and doesn’t do a good job of explaining why you performed the study. It feels like a random dump of information with no clear objective for the experiments performed. I am an expert in the field and I still cannot make sense of the introduction.
  • Data presented in section 3.1 should be presented after 3.2.

Reviewer 3 Report

The authors have properly addressed all my concerns